# Advances in Biosynthesis and Biological Functions of Proanthocyanidins in Horticultural Plants

**DOI:** 10.3390/foods9121774

**Published:** 2020-11-30

**Authors:** Dan Yu, Ting Huang, Bin Tian, Jicheng Zhan

**Affiliations:** 1Beijing Key Laboratory of Viticulture and Enology, College of Food Science and Nutritional Engineering, China Agricultural University, Beijing 100085, China; yudiane@cau.edu.cn (D.Y.); b20173060474@cau.edu.cn (T.H.); 2Faculty of Agriculture and Life Sciences, Lincoln University, Lincoln 7647, New Zealand

**Keywords:** abiotic stress, biotic stress, proanthocyanidins, regulatory mechanism

## Abstract

Proanthocyanidins are colorless flavonoid polymers condensed from flavan-3-ol units. They are essential secondary plant metabolites that contribute to the nutritional value and sensory quality of many fruits and the related processed products. Mounting evidence has shown that the accumulation of proanthocyanidins is associated with the resistance of plants against a broad spectrum of abiotic and biotic stress conditions. The biosynthesis of proanthocyanidins has been examined extensively, allowing for identifying and characterizing the key regulators controlling the biosynthetic pathway in many plants. New findings revealed that these specific regulators were involved in the proanthocyanidins biosynthetic network in response to various environmental conditions. This paper reviews the current knowledge regarding the control of key regulators in the underlying proanthocyanidins biosynthetic and molecular mechanisms in response to environmental stress. Furthermore, it discusses the directions for future research on the metabolic engineering of proanthocyanidins production to improve food and fruit crop quality.

## 1. Introduction

Horticultural crops, including vegetables and fruits, are a crucial food source for human nutrition as they contain carbohydrates, proteins, vitamins, organic acids, and minerals. Proanthocyanidins (PAs), also known as condensed tannins, are among the most abundant polyphenols in plants. They are naturally present in woody plants and herbaceous species. PAs are present in the leaves, fruits, seeds, roots, and other parts of plants [1], and play essential roles in the growth and development of leaves and fruits, as well as, the modulation of seed dormancy and germination [2,3,4,5,6,7]. In addition, PAs also influence mouthfeel by enhancing the astringency of fruits and in their processed products, e.g., wines and beverages [8,9]. Moreover, as potential dietary antioxidants, PAs are considered beneficial to human health. It has been widely accepted that the PAs from food plants and medicinal plants are associated with various bioactivities, displaying immunomodulatory, anti-inflammatory, anti-cancer, anti-microbial, and hypolipidemic properties while reducing the risk of cardiovascular disease and ameliorating obesity [10,11,12,13,14,15,16,17,18,19,20,21,22].

Developmental regulation is considered the driving factor for PAs biosynthesis, which can also be influenced by environmental fluctuations. In grape berry, the PAs accumulate in the skin prior to véraison, after which the total PAs content decreases due to intermediate metabolites deviation [4]. Many crops have shown a similar decreasing trend in PAs levels during the fruit ripening process, including bilberries, persimmons, and blueberries [23,24,25]. PAs contribute to the astringency and bitterness of the young fruits, preventing them from being eaten before they are ripe [26]. The accumulation of PAs is associated with the resistance of plants against various biotic and abiotic stimuli, such as low temperature [27,28], drought [29], wounding [30,31], UV radiation [30], and fungal pathogens [32,33,34]. Therefore, increasing attention has been focused on the metabolic engineering and regulatory mechanism of PAs in horticultural plants. The structural genes and key factors that regulate PAs biosynthesis were identified and characterized in many plants, particularly in horticultural crops rich in PAs, such as grapevines, poplars, and apples. Studies involving PAs biosynthesis regulation in response to environmental stresses revealed that the regulatory network was based on the regulators belonging to the MYB, MYC-like basic helix-loop-helix (bHLH), WD40-repeat proteins, and the MYB-bHLH-WD40 (MBW) complexes. However, research regarding more specific regulators in this complicated network is required to understand the interaction between these vital regulators during PAs biosynthesis to ultimately improve the quality of crops.

## 2. Structure and Biosynthesis of Proanthocyanidins

PAs consist of a mixture of flavan-3-ol units and flavan-3,4-diols (leucoanthocyanidins) in complicated ways [35]. Flavan-3-ol monomeric units include (+)-catechins, (−)-epicatechins, and their derivatives: (+)-gallocatechin, (+)-catechin gallate, (+)-gallocatechin gallate (−)-epicatechin gallate, (−)-epigallocatechin, and (−)-epigallocatechin gallate (Figure 1). Both (+)-catechins and (−)-epicatechins are believed to act as the starter units for PAs polymerization, and flavan-3-ols and flavan-3,4-diols serve as extension units for further PAs extension [1,36,37]. Different numbers of subunits polymerized to form oligomeric or polymeric PAs [1,38]. The degree of polymerization of PAs varies among species and tissues. Based on the interflavanic linkages, the subunits linked by C4-C8 and/or C4-C6 bonds are known as B-type PAs, and those with additional C2-O-C7 or C2-O-C5 bond in the structure are classified into A-type PAs (Figure 1) [1]. In addition, esterification, glycosylation, and other modifications have also been found in PAs with more complex structures. B-type PAs are widely distributed in plants and foods, such as *senna alata* leaves, pine trees, black wattle, beans, barley, sorghum, seeds, fruits, berries, nuts, cinnamon, chocolate, and wine [39,40,41,42,43,44,45]. A-type PAs are mainly found in bilberry, peanuts, plums, cranberries, curry, and cinnamon [46,47,48,49].

The pathway of flavonoid biosynthesis has been well studied in various plants (Figure 2). Flavonols, anthocyanins, and PAs are the main products of the flavonoid biosynthetic pathway, which belongs to the plant secondary metabolism [50,51,52]. They share the common upstream steps in the biosynthetic pathway of flavonoids, and their precursor is dihydroflavonols which can be converted to leucoanthocyanidins by dihydroflavonol-4-reductase (DFR) and to flavonols by flavonol synthase (FLS). Leucoanthocyanidins can be further catalyzed to produce anthocyanidins by the oxygenation reaction of anthocyanidin synthase (ANS) or leucoanthocyanidin dioxygenase (LDOX). Leucoanthocyanidins and anthocyanidins can be catalyzed by leucoanthocyanidin reductase (LAR) and anthocyanidin reductase (ANR) to produce 2,3-*trans*-flavan-3-ols (such as (+)-catechin) and 2,3-*cis*-flavan-3-ols (such as (−)-epicatechin) which are the key subunits involved in the last steps of PAs biosynthesis.

To date, *LAR* and *ANR* genes have been cloned from several plant species, and their functions have been demonstrated both genetically and biochemically [3,36,53,54,55,56,57,58,59,60]. As is well known, leucoanthocyanidins are converted to (+)-catechins through the catalysis of LAR. Interestingly, ectopic expression of tea *CsLAR*, and cacao *TcLAR* in tobacco showed that the production of (−)-epicatechin was higher than that of catechins [57,61]. More recent studies also found that LAR from *Medicago truncatula* and grapevine can convert 4β-(S-cysteinyl)-epicatechin into (−)-epicatechins, thereby regulating the degree of PAs polymerization [59,62]. In addition to catalyzing the synthesis of (−)-epicatechins from anthocyanidins, ANR is proven to convert anthocyanidins to a mixture of (−)-epicatechins and (+)-catechins [61,63]. This suggests that LAR and ANR may possess more features to be explored [61,63,64].

Most of the enzymes for flavonoid biosynthesis are located on the endoplasmic reticulum and tonoplast of cells. PAs are synthesized in the cytoplasm, and then transferred and accumulated into the vacuole and apoplast [65,66,67]. In seed, PAs are mostly accumulated in the inner endothelial layer of seed testa [35,68,69], and the brown color of the seed coat may result from the oxidation of PAs [70]. The seeds without PAs content are called *transparent testa (tt)* phenotype and *tannin deficient seeds*, which exhibit yellow or pale brown in color [54,71,72]. The PA-deficient mutants have become an ideal material for investigating the biosynthesis and transportation of PAs. MATE (multidrug and toxic compound extrusion) and GST (glutathione S-transferase) are proteins responsible for flavonoids transportation. AtTT12, a member of the MATE family, localizes the PAs in the vacuoles of the testa of Arabidopsis seeds [73]. *AtTT13/AtAHA10* encodes a protein belonging to the ATPase family, which is required for the deposition of PA precursors in vacuole and functions as a proton pump, providing the driving force for TT12-mediated transportation of PA precursors [74]. VvMATE1 and VvMATE2 are transport proteins located in the tonoplast and Golgi complex, respectively, and involved in the transport and accumulation of PAs through different ways in grapevine [67]. GST proteins were thought to deliver anthocyanins into vacuoles, e.g., ZmBz2 in maize, PhAN9 in petunia, AtTT19 in Arabidopsis, and VviGSTs in grape [37,75,76]. After heterologous overexpression of *PhAN9* in *Arabidopsis tt19* mutant, the anthocyanin pigment in seedlings was complemented, but no complementation was observed in the pale brown seeds of the *tt19* mutant [37]. Moreover, PAs precursors were dispersed in the cytoplasm in *tt12* mutant [73], but accumulated as membrane-like structures in endothelium cells of *tt19*, indicating that TT19 may function as a subsequent transporter to deliver the vesicles containing PAs precursors to the central vacuole [37,77]. VviGST3, a homolog to AtTT19, was highly expressed in the seed during grape berry development, and the complementation in seed color was observed in VviGST3 overexpressed *tt19* mutant, suggesting that VviGST3 may be associated with the transport of PAs [78]. In addition, heterologous expression of pear *PcGSTF12* in the *Arabidopsis tt19* mutant showed that *PcGSTF12* did not complement the mutant seed color to the normal brown, but it promoted the procyanidin A3 accumulation in the mutant and affected the transcription of PAs- and anthocyanin-related genes. The results of this study indicated that PcGSTF12 may be involved in the transport and accumulation of PAs and anthocyanins [79].

The vacuole is the final destination for the transportation of PAs, and the polymerization of PAs probably takes place here. Although little is known about the polymerization mechanisms of PAs, the enzymatic and nonenzymatic polymerization reactions are widely discussed [1,80,81]. Plant polyphenol oxidases, TT10 and plant peroxidases may take part in PAs polymerization and oxidation [1,65,82,83]. For instance, *AtTT10* encodes a putative laccase that oxidizes epicatechin into yellow or brown oligomers, which may involve in PAs oxidative browning in the seed testa [65]. 

## 3. Transcriptional Activator of Proanthocyanidins Biosynthesis

The biosynthesis rate of PAs is mainly controlled by the key enzymes within the pathway, and these enzymes are regulated by specific transcriptional regulatory factors (TFs) and regulatory proteins. In the past few years, several TFs and proteins that either directly or indirectly regulate PAs biosynthesis have been identified from model plant and horticultural crops. The regulation involves different MYBs, bHLHs, and WD40 proteins, and many other key factors. The MYB and bHLH TFs are found in all eukaryotes, representing one of the largest plant transcription factor families according to the number of members. They play a pivotal role in growth and development, hormone signal transduction, and plants’ stress responses [84,85,86,87]. Depending on the structure of the DNA binding domain, the MYB TFs are classified into four subfamilies: 1R-MYB, 2R-MYB, 3R-MYB, and 4R-MYB. The R2R3 MYBs, belonging to 2R-MYB subfamily, are most widely involved in the regulation of the flavonoid pathway, followed by the R3 MYB of 1R-MYB subfamily [85,88]. 

MYB with [(D/E)LX2(R-K)X3LX6LX3R] motif could be able to combine with the MYB-interacting region (MIR) at the N-terminal of bHLH [87], and MYB and bHLH can also bind to the downstream target gene promoters, respectively. It has been confirmed that MYB family members can recognize the MBSI (CNGTTR3) region and AC-rich ([A/C]CC[A/T]A[A/C]) region, while bHLH proteins specifically bind to E-box (CANNTG) or G-box (CACTG) cis-elements [89]. The MYB protein is often considered to be as a core of MBW complexes in determining DNA binding specificity. The bHLH participates in histone modification and affects the specificity of DNA binding [90], and functions as the bridge between MYB and WD40 protein to form a stable ternary protein complex. WD40 is a type of protein composed of multiple WD motifs in tandem. It often combines with other members of the family or other proteins to modulate target genes to regulate the flavonoids synthesis, signal transduction, and cell processes, such as cell division, vesicle formation and transport, packaging and processing [91,92]. The activity of the MYB-bHLH complex depends on the expression of TTG1 (WD40) in Arabidopsis protoplasts [93], indicating the WD40 protein is crucial for the stability of the MBW complexes. The complexes control the synthesis of flavonoids, including PAs, through the activation of structural gene expressions by binding with cis-elements of their promoters [87,88,90,93].

The MBW complex in Arabidopsis, composed of AtMYB123 (TT2), AtbHLH42 (TT8), and TTG1 proteins, was reported to activate the expression of *DFR*, *LDOX*, and *ANR (BAN)* genes, which lead to the accumulation of PAs in the seed coat [93,94]. A large number of homologs of TT2, TT8, and TTG1 have been identified in various plants, such as FaMYB9/11, FabHLH3, and FaTTG1 in strawberry [95], LjTT2a/2b/2c in *Lotus japonicus* [96], VvMYBPA2/PAR in *Vitis vinifera* [97], TaMYB14 in *Trifolium arvense* [98], TaMYB10-A1/-B1/-D1 in wheat [99], HvMYB10 in *Hordeum vulgare* [100], Tc-MYBPA in *Theobroma cacao* [101], MtMYB5/14 in *M. truncatula* [102], MdMYB9/11/12 in apple [27], PtMYB115 in poplar [34], CsMYB60 in cucumber [103,104], RrMYB10 in *Rosa rugosa* [31], AabHLH1 in *Anthurium andraeanum* [105], DkMYB2/4, DkMYC1, and DkWDR1 in persimmon [24,106,107], MtWD40-1 in *M. truncatula* [108], FhTTG1 in *Freesia hybrida* [109], and SlAN11 in tomato [110]. These factors are transcriptional activators of PAs biosynthetic pathways, suggesting that the MBW complexes are well conserved in higher plants [87,94]. The transcription of *AtTT8* and *CsNoemi* (bHLH) was driven by an overexpressed MYB protein (AtTT2 and CsPH4) in the transgenic plants [94,111]. These results suggested that certain MYBs not only combine with bHLH to regulate the downstream of structural genes but also act as a regulator of *bHLH* to form a positive feedback loop [77]. In addition to TT2-MYB, there is a specific TFs, PA1-type MYB, to regulate PAs biosynthesis. MYBPA1 was first reported in grape, of which homologs were subsequently identified in persimmon [24], nectarine [112], poplar [113], and apple [27]. Recently, a new MYB family member, PpMYB7, was identified in peach. It can bind to the promoter of *PpLAR*, but not *PpANR*, to regulate PAs synthesis [114]. *MtPAR* is a novel MYB TF found in the seed coat of *M. truncatula*, which enhanced the PAs accumulation by regulating another positive regulator *MtWD40-1*, but not the *ANR* [115]. It has been reported that many MYBs require a bHLH partner to activate the promoter of downstream genes [4]. For instance, VvMYB5a and VvMYB5b were able to activate the promoters of flavonoid-related genes (e.g., *VvCHI* and *VvLAR1*) at the presence of AtEGL3, a bHLH protein involved in flavonoid pathway regulation, and thus elevated accumulation of PAs in the ectopically overexpressed tobacco [116,117]. VvMYC1, a bHLH transcription factor of grapevine, can interact with VvMYBPA1 to form a complex which regulates the transcription of two PAs structural genes (*CHI* and *ANR*) and the accumulation of PAs in skins and seeds of berries, but VvMYC1 alone is not able to promote the expression of these two PAs structural genes [118]. 

MBW is not the only regulator that regulate flavonoid biosynthesis. There are several other types/families of TFs which can regulate PAs biosynthetic pathway in plants. A recent study reported that the apple NAC52 can bind to the promoters of *MdMYB9*, *MdMYB11*, and *MdLAR*, and promote PAs accumulation in the *MdNAC52*-overexpressed calli [119]. AtTTG2 and VvWRKY26 belong to the WRKY family. AtTTG2 was shown to be responsible for PAs formation in seed coat by directly regulating *TT13* and *TT12* [120]. VvWRKY26 from *V. vinifera* was identified as a key regulator that controls the vacuolar transportation and the accumulation of flavonoids, especially the PAs in the berries [121]. Arabidopsis TT16, an ARABIDOPSIS BSISTER (ABS) MADS domain protein, was required for *BANYULS (ANR)* transcript and the deposition of PAs in the endothelium of the seed coat [122]. The *Brassica napus* TT16 homologs, BnTT16, regulates the transcriptional abundance of PA-related genes (*CHS*, *CHI*, *F3H*, *LDOX,* and *TT10*). Furthermore, a reduction was evident in the PAs accumulation in the *Bntt16* RNAi seeds [123]. The PAs transcriptional regulatory networks are complex but important to investigate the PAs metabolism in plants.

## 4. Transcriptional Repressors Regulate Proanthocyanidins Synthesis

Feedback regulation mechanisms also play an important role in the regulation of PAs biosynthesis in plants. Despite the crucial role of transcriptional activator in PAs biosynthesis, some proteins interfered with the accumulation of PAs have been studied as well. Previous reports have documented that strawberry FaMYB1 not only negatively regulated anthocyanin and flavanol biosynthesis in *Fragaria* and tobacco [124], but also specifically inhibited the accumulation of PAs in the leaves of *Lotus corniculatus* [125]. Three negative regulators of PAs were identified in grapevine, namely, VvMYBC2-L1, VvMYBC2-L2, and VvMYBC2-L3. They have high homology with FaMYB1 and anthocyanin repressor PhMYB27 in petunia. It has been shown that VvMYBC2-L1 interacted with VvMYC1 and PhAN1 (a bHLH protein involved in anthocyanin biosynthesis in petunia), and *VvMYBC2-L2* and *VvMYBC2-L3* had similar expression profiles in grape tissues [126], indicating that they may possess similar functions. Overexpression of VvMYBC2-L1 and VvMYBC2-L3 in grapevine hairy roots lead to a significant reduction in PAs, and VvMYBC2-L1 and VvMYBC2-L3 also decrease both anthocyanin levels in flower petals and the PAs in the seeds of the transgenic petunia lines [126,127]. In previous reports, a few of MYB TFs were upregulated in *PtMYB134-* and *PtMYB115*- overexpressed poplar seedlings [113]. Surprisingly, among them, PtMYB182, PtMYB165, and PtMYB194 were characterized as R2R3 MYB repressors in further studies. These three negative regulators reduced the synthesis of PAs and anthocyanins in transgenic seedlings/tissues, and suppressed the transcription of several flavonoid-/phenylpropanoid-related genes. Moreover, the repressors competed with MYB activators for binding to bHLH co-activator and disrupted the transcriptional activation of MBW complex [128,129]. Similar R2R3 MYB repressors were detected in *M. trunculata* and peaches. *M. trunculata MtMYB2* and peach *PpMYB18*, which can be induced by the PAs activators, inhibit the biosynthesis of PAs and anthocyanin in transgenic plant seeds and tissues [130,131]. In a dosage-response repression assay, a small dosage of *PpMYB18* was enough to reduce the activity of activator (PpMYB10.1), indicating that PpMYB18 acts as a passive repressor and no titration effect was observed in the transient assays when the dosage ratio of *PpMYB10.1* to *PpMYB18* increased above 1:1, suggesting that PpMYB18 functions as a passive repressor by interacting with bHLH cofactor, which inhibits the transcriptional activity of the MBW complex [131]. Several studies have reported that the JAZ protein negatively regulates the flavonoid pathway by interacting with the MBW complex [132,133]. More recently, tartary buckwheat FtMYB18 has also been proposed as a negative regulator of PAs and anthocyanin pathway. Evidence from transgenic plants/hairy roots has shown that the FtMYB18 suppressed the accumulation of PAs and anthocyanin by inhibiting the expression of genes related to flavonoids. The inhibition of FtMYB18 was achieved by combining with FtTT8 and FtTTG1, or forming a MYB-JAZ complex with FtJAZ1/-3/-4/-7 protein [134]. As described above, the MYB inhibitor regulated the metabolism of multiple flavonoids including PAs. Therefore, MYB inhibitors seemed to have a less specificity than that of MYB activators, which may be determined by its binding properties to bHLH [135]. In addition, AtMYBL2, a repressor belonging to R3-MYB family in *Arabidopsis*, was identified to function as a negative regulator of flavonoid pathway. It inhibited the activity of MBW complex by binding with TT8 and GL3 to control the flavonoid structural gene expression, thus reducing the accumulation of PAs in seed [136]. 

The inhibitor from the other TFs family has also been identified. The expression of *LcbHLH92* in sheepgrass appeared to be negatively correlated with the anthocyanin/PAs-related genes. Overexpression of *LcbHLH92* in *Arabidopsis*, the transcript of *DFR* and *ANS* were downregulated, and the synthesis of anthocyanin and PAs were inhibited. Electrophoretic mobility shift assay (EMSA) and chromatin immunoprecipitation and quantitative real-time (ChIP-qPCR) assay have shown that the LcbHLH92 can bind to the promoter of *JAZ* and JAZ inhibits the expression of *TT8*, resulting in a decrease in the biosynthesis of flavonoid [7]. The bric-à-brac/tramtrack/broad complex (BTB) protein is a component of the CUL3-RBX1-BTB protein E3 ubiquitin ligases complex, which mediated the degradation of diverse protein [137]. The BTB protein has been found to take part in plant growth and development and responding to various stresses [138,139]. Recently, MdMYB9 protein was shown to be degraded by MdBT2 via the 26S proteasome system, resulting in a substantial reduction of anthocyanin and PAs in apple [140].

The same as PAs activators, the repressors are an important part of transcription regulation as well. They are often regulated by other transcription factors and signals to avoid excessive accumulation of PAs in plants [125,127,135,136]. Therefore, greater insights into the mechanisms of negative regulators will help us better understand the PAs biosynthesis in response to environmental stresses.

## 5. Proanthocyanidins in Response to Abiotic Stresses

Plants frequently suffer from a range of environmental stresses, such as low temperature, high light intensity, drought, wounding, and other abiotic stressors. Extreme environmental conditions could disrupt the natural defense system, and the severe stresses will provoke the production and accumulation of reactive oxygen species (ROS) and the induction of oxidative stress responses in plant cells, thereby hindering metabolic activities and affecting the integrity of organelles of plants, which limits the productivity of crops [141]. In response to unfavorable environmental conditions, plants have evolved excellent defense mechanisms to perceive and adapt to these external stresses. Flavonoids, a group of protective phenolic substances of plants, can be induced significantly by the stressful conditions, and participate in response to adversity as antioxidants or as signal molecules [142,143,144,145].

The biosynthesis of PAs in response to multiple abiotic stress and the underlying mechanism has been elucidated in numerous studies. Previous research shows that, compared with the control, cucumber seedlings pretreated with PAs had less oxidative damage induced by high irradiance, artificial drought, and cold stress [146]. In tartary buckwheat, RNA-sequencing and GC-TOF MS analysis found that low temperature upregulated relevant genes of flavonoids metabolism and triggered the accumulation of PAs monomers (catechin and epicatechin), implying that PAs might be associated with enhanced tolerance to cold stress [147]. Nevertheless, how the PAs in plants respond to high temperatures is elusive. Some studies reported that high temperature did not affect PA content in grape skins during the growing season [148,149], but other studies noticed that high temperature reduced the PAs production in grape berries [150,151], and decreased in *VvANR* and *VvLAR1* mRNA levels [151]. Researchers noticed that water deficit could promote the secondary metabolism in grape berries. However, a study showed that both regulated deficit irrigation and non-irrigation upregulated the expression of PAs-related genes in grape seeds, but did not lead to changes in PAs concentration, which may be due to other factors such as oxidation and/or degradation of PAs in the last stage of ripening [152]. As grape berry skin is exposed to the external surroundings directly, the biosynthesis of PAs in grape skin seems to be more susceptible to temperature than those in the seed [153]. Water deficit could induce the expression of *VvLAR2* and *VvMYBPA1,* and the concentration and polymerization of PAs in Cabernet Sauvignon grape skins. High sunlight exposure has been shown to boost the PAs synthesis in *Cistus clusii* [154], poplar [30,155], larch [156], and apple [157]. The excess sunlight exposure increased the concentration of PAs in *Cistus clusii*, which was more obvious in older plants because they suffered more from UV light stress [154]. Transcriptome sequencing analysis showed that genes related to flavonoid biosynthesis were up-regulated in *Pohlia nutans* under high salt stress. Moreover, pre-treatment with 1% PAs significantly improved the survival rate of moss [158]. 

It has been reported that certain TFs can regulate the PAs biosynthesis in different manners in response to abiotic stresses. The transcriptional level of *LoMYB29* in *Larix olgensis* was upregulated by wounding, high light intensity, NaCl, PEG6000, methyl jasmonate (MeJA), and abscisic acid (ABA) treatments, and ectopic overexpression of *LoMYB29* enhanced the PAs to accumulate in *Arabidopsis* leaves [159]. This is in agreement that wounding, oxidative stress and salicylic acid (SA) can trigger the upregulated expression of *RrMYB5*, *RrMYB10* (*Rosa rugosa*), and *RcMYBPA2* (*Rosa chinensis*), which can then activate the promoter of PAs structural genes and result in accumulation of PAs in roses [31,160]. PAs are known as a powerful antioxidant that has been proved to increase the antioxidant capacity of plants [6]. Overexpression of *RrMYB5* and *RrMYB10* in rose somatic embryos and seedlings, not only increased PAs content, but also enhanced the activity of catalase, peroxidase and superoxide dismutase of plants/tissues to low accumulation MDA and scavenge ROS, which improved tolerance to the stresses [31,160,161]. This means that PAs improve plant tolerance to environmental stresses by regulating the antioxidant system to facilitate ROS scavenging. In addition, the production of endogenous ABA was promoted in the rose with over-production of PAs, and the ABA signaling pathway was closely related to the response to stress in plants [31,161]. Wounding, high sunlight exposure, and nitrogen deficiency caused oxidative stress in poplars, inducing the accumulation of PAs in the leaves and significantly upregulating the expression of *PtMYB134* and *PtMYB115* [30,34,155,162]. These MYB factors overexpressor in poplar seedlings exhibited dramatically higher PAs concentration, which reduced the level of H_2_O_2_, and showed the potential function in reducing leaf necrosis and photosystem II by the ROS damage [155]. *MdMYB9/11/12/23* and *MdMYBPA1*, which can bind to the promoters of PAs structural genes, were positive regulatory factors of PAs synthesis in apple, and could also be triggered in apple by low temperature. When constitutively expressed in apple calli, the factors induced the transcription of PAs synthesis, elevated levels of the PAs to reduce the oxidant damage caused by cold stress [27,28,157]. Furthermore, a recent study found that MdMYB23 protein was induced by cold stress, but degraded via the ubiquitin-proteasome proteolytic pathway in normal conditions. MdBT2 decreased PAs accumulation by promoting the degradation of MdMYB23 [28], which can balance the levels of PAs in apple.

In general, uncomfortable temperature, high light exposure, drought, wounding, nitrogen deficiency, and other stressful conditions would trigger the generation of ROS and cause oxidative damage to plants. Plants have developed an antioxidant defense system to manage oxidative damage. Like other flavonoids, PAs possess an ortho-hydroxylated B-ring, which is a key function of phenolic antioxidants [163,164]. PAs exhibit a strong antioxidant capacity in vitro. However, the ROS are mainly generated in chloroplasts and mitochondria and are short-lived. By contrast, H_2_O_2_ is more stable and can cross membranes and diffuse into the vacuole [165,166]. Therefore, PAs may have an important role in the H_2_O_2_ scavenging system. However, the specific process and mechanism are still unclear. External stimulation promotes the PAs accumulation by activating plant hormones and MYB TFs (TT2- and PA1-type). The synthesis of PAs in plant cells is considered to be an adaptive response to oxidative stress [155]. As outlined above, plants containing high PAs levels display excellent antioxidant properties that can remove excess ROS and alleviate further damage to plants, improving their ability to respond to various stimuli [31,33].

## 6. Proanthocyanidins in Response to Biotic Stresses

In addition to abiotic stresses, plants often encounter biotic stresses, such as fungi, bacteria, viruses, and insects, which are highly contagious. These biotic stresses can cause the death and decay of plants, and consequently affect the quality and yield of crops. Chemical pesticides provide a broad-spectrum of resistance to pests and diseases, and it is the most effective and commonly used method. However, the long-term and improper use of chemical pesticides can cause the resistance evolution of pathogens and pests to the chemicals, negatively influencing the environment and food safety. Previous studies have shown that PAs are substantially involved in plant resistance against biotic stress, such as fungal infection and insect herbivores [1,32,33]. PAs can be considered to be potential biosafety and eco-friendly strategy to protect plants against biotic stresses.

As a kind of pre-formed protective barrier, PAs are distributed in the endothelial layer of the seed coat in many plants to protect the seed from biotic stresses [1]. PAs are also found to localize in the epidermis and vascular bundles of poplar leaves. After inoculation with rust fungus (*Melampsora larici-populina*), more PAs accumulated on the site of fungal infection in the epidermis of leaves, indicating that the location of PAs provided a defense barrier for leaves to resist early colonization of the fungus [33]. Using transcriptome sequencing analysis, it was found that, after fungus (*Melampsora medusae*) inoculation, the PAs-related genes and the accumulation of PAs in poplar leaves were strongly induced [167]. At least 2–3 times higher levels of catechin and PAs were observed in the leaves of poplar genotype with moderate resistance against rust infection compared to the ones sensitive to rust infection, and there was an increase in the concentration of flavan-3-ols in response to fungal infection. In addition, there was a negative correlation between the degree of disease occurrence and the PAs concentration of leaves [33]. Furthermore, PAs possessed toxic and growth inhibitory effects on microbial pathogens [168]. Merlot grapevine treated with benzothiadiazole (BTH) resulted in a 36% of the increase in PAs and reduced the incidence and severity of gray mold caused by *Botrytis cinereal* [169]. In bilberry the biosynthesis and accumulation of PAs were significantly increased when the berry was infected by endophyte (*Paraphaeosphaeria* sp.) and *Botrytis cinereal*, and the synthesis of epigallocatechin was greatly induced due to the infection of *Botrytis cinereal* [170]. When inoculating fungus on strawberries with different maturity levels, the riper strawberries were much more susceptible to fungus than the less ripe ones as fungal development in unripe fruits was inconspicuous and the expression levels of *LAR* and *ANR* were higher in unripe fruits [171,172]. Meanwhile, the concentrations of flavan-3-ols and PAs have been found to decrease during the ripening stages [173,174], indicating that flavan-3-ols and PAs derivatives were considered to be pre-formed defense mechanisms of fruit by inhibiting the fungal invasion. Constitutive expression of *PtLAR3* in poplar results in a boost to the level of PAs and enhanced the ability to resistance to the fungus *Marssonina brunnea*. Compared with wild-type ones, the crude polyphenol extract from transgenic leaves inhibited the growth of fungus more effectively and has a toxic effect on mycelium which showed abnormal growth [32]. Previous research indicated that PtMYB115 and PtMYB134 interacted with PtbHLH131 to modulate PAs synthesis in poplars. In greenhouse-grown transgenic of *PtMYB115* poplar seedlings, the transcriptions of PAs biosynthesis gene were activated, which increased PAs concentration and fungus resistance. However, the CRISPR/Cas9-generated *myb115* mutant, which has lower PAs levels and decreased expressions of PAs-related genes, exhibited high sensitivity to fungus [34]. Similarly, overexpression and silencing of *PtMYB134* also affected the content of PAs in poplar, which in turn affected resistance to rust fungus [33]. These findings further demonstrate that PAs are effective antifungal materials and play a critical role in the resistance of plants to fungus.

It has been proven that insect herbivores often induce PA accumulation in plants. For example, under natural conditions, in the leaves of old-growth black poplar, the low molecular weight flavan-3-ols and PAs concentration were not affected by gypsy moth (*Lymantria dispar*) larva feeding. However, in the single tree experiment, after caterpillar infestation, the PAs in the leaves increased by 10–20% and the production of low molecular weight flavan-3-ols decreased by 10% in the leaves but increased by 10% in the bark [175]. The forest tent caterpillar (*Malacosoma disstria*) and satin moth (*Leucoma salicis*) larva could strongly increase the expression level of the *DFR* gene, resulting in higher production of PAs in the poplar leaves [162]. Heterologous expression of tea *CsDFR* and *CsANR* in tobacco increased the generation of flavan-3-ols (catechin, epicatechin, and epigallocatechin) and protective abilities of plant anti-herbivores [176]. PAs have a deterrent and/or toxic effect on herbivorous insects, which may be one of the pre-formed protective barriers in plants [177,178]. However, there is also evidence that PAs did not possess broad-spectrum antiherbivorous activity. Researchers found that, compared with other low doses of plant metabolites, the PAs in high dose had little success against the insects [179]. Therefore, PAs have been shown to contribute to plant defense against a wide range of fungal and bacterial pathogens by its physical and biochemical properties. This can be considered one of the best methods of comprehensively guiding agricultural production, and preventing pathogen and insect invasion.

## 7. Hormones Regulate Proanthocyanidins Synthesis

A growing body of evidences illustrated that the plant hormones are closely involved in PAs synthesis. The phytohormone can interact with the MBW complex related to flavonoid synthesis at transcriptional or post-transcriptional levels [180]. Pre-treatment of grape berry with ABA can reduce the level of PAs accumulation by inhibiting the activity of LAR and ANR, and the expression of PAs-related genes [181]. EMSA and transient reporter assay showed that DkbZIP5, a TF involved in ABA signaling pathway, is bound to ABA-responsive elements in the *DkMYB4* promoter, and thus ABA can signal the regulation of PAs biosynthesis in persimmon by regulating the expression of *DkMYB4* through *DkbZIP5* [182]. Exogenous application of MeJA has been shown to increase PAs content in grape berries [183]. A study on apple suggested that jasmonic acid (JA) stimulated PAs accumulation by up-regulated *MdMYB9* and *MdMYB11* [184]. Detailed study showed that MdSnRK1.1 can phosphorylate MdJAZ18, which acts as a repressor in the jasmonic acid (JA) signaling pathway. This interaction ultimately mediates the degradation of MdJAZ18 via the 26S proteasome, releasing MdbHLH3, the co-activator of MdMYB9 and MdMYB11, while promoting the PAs and anthocyanin production [185]. Two transcription factors belonging to the ethylene response factors (ERF) family of *Malus Crabapple*, *RAP2-4*, and *RAV1*, specifically bound to the promoter of *McLAR1* and *McANR2*, respectively, and acted as positive- (*RAP2-4*) and negative- (*RAV1*) regulators that control PAs biosynthesis. Furthermore, plant hormones are involved in plant resistance to biotic and abiotic stresses [186]. 

Plant hormones have an important function in plant-pathogen interaction [187]. Generally, SA is mainly involved in plant resistance to biotrophic pathogens and piercing–sucking insects, and JA is mainly involved in resistance to necrotrophs and chewing insects [188]. Foliar rust fungi promoted the increase of SA, JA, and ABA in infected poplar leaves, but only the rising trend of SA was coordinated with that of flavan-3-ol. Exogenous application of benzothiadiazole, SA analogue, increases the expression of PAs-related MBW complex and the concentration of flavan-3-ol, and reduces the rust fungus invasion [189]. Likewise, another study in strawberry reported the same result. When *Podosphaera aphanis* infected strawberry leaves, SA-related and phenylpropanoid pathway-related genes were activated, while JA pathway-related genes were down-regulated. Moreover, SA can induce the transcription of PAs-related MBW complex and PAs accumulation in leaves, and consequently inhibits the pathogen growth [190]. In Norway spruce, SA supplement induces the biosynthesis of PAs, thereby reducing the damage from bark beetle [191]. The black poplar stems infected with *Plectosphaerella populi* showed increased content of flavan-3-ols, SA, and cytokinins (CKs), but the levels of JA and JA-isoleucine (JA-Ile) were only raised at the early stage of infection. Exogenous application of CKs resulted in the decrease in SA and flavan-3-ols concentrations. These results suggested that SA is the primary hormone that control antifungal flavan-3-ols and PAs synthesis [192]. 

The plant hormones as signals may link the changing environmental conditions and development cues and PAs biosynthesis. Although many studies illustrated the links between plant hormones and PAs synthesis, the relationship between plant hormones and the TFs regulating PAs biosynthetic pathways requires further investigation. 

## 8. Conclusions and Further Prospects

PAs have attracted attention not only because they possess beneficial pharmacological properties but also because of their special role in the regulation of fruit quality and plant defense. The principal focus of research in the biosynthetic pathway and regulatory mechanism of PAs has been explored for years, and some encouraging progress has been made. However, there are still many issues that need to be further explored. The external environment in which plants live is constantly changing, there is a diversity of species, and the complex genetic background of woody plants, results in different regulatory mechanisms of PAs in various species in response to developmental cues and various stresses (Figure 3). In all species analyzed to date, a large number of MYBs regulate the synthesis of PAs that are involved in plant response to stress and signals. Nevertheless, the bHLH and WD40 cofactors of the MBW complex are also involved in the regulation of other pathways, so their specificity for PAs-regulation is less than that of MYB TFs, and few studies have focused on bHLH TFs, WD40 protein, and the proteins from other families that also participate in the biosynthesis of PAs. The highly induced transcription and accumulation of PAs are considered to be a systemic response of plant to adverse conditions [162]. Further work needs to reveal how the biotic and abiotic stresses trigger the transcription of these TFs, how the transcriptional and post-transcriptional regulation modifies these TFs, and how the crosstalk between these TFs and various signals and hormones is conducted. 

Metabolic engineering and molecular biology methods can be used to produce PA-rich crops that can enhance the tolerance of crops to various environmental stress, and improve the quality and nutritional value of crops. PAs have a great impact on the sensory properties of fruits by contributing to the astringency and bitterness, and may also influence the digestive system in humans and animals [1,193]. For example, the accumulation of PAs in persimmon fruit leads to astringency [194], which brings unpleasant tactile sensation. Understanding the regulation of PAs biosynthesis network can provide further insights into more targeted gene modifications, and are of great importance for breeding. However, it should be noted that one factor might affect the biosynthesis of many other metabolites. For example, the overexpression of *VvMYBPA1* in tobacco upregulates PAs metabolism while downregulating anthocyanin biosynthesis [195]. Therefore, the comprehensive and in-depth analysis of the molecular regulation mechanism of plant resistance mediated by PAs has a positive significance for improving our knowledge of regulatory network controlling PAs synthesis and provides a theoretical foundation for genetic improvement and breeding of the plants with moderate levels of PAs. 

## Figures and Tables

**Figure 1 foods-09-01774-f001:**
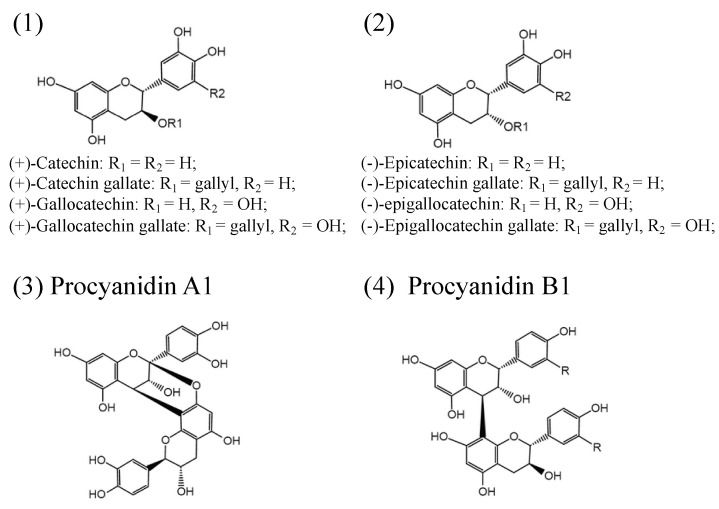
Structures of the flavan-3-ols (**1**,**2**); simple A1-type proanthocyanidins (**3**); simple B1-type proanthocyanidins (**4**).

**Figure 2 foods-09-01774-f002:**
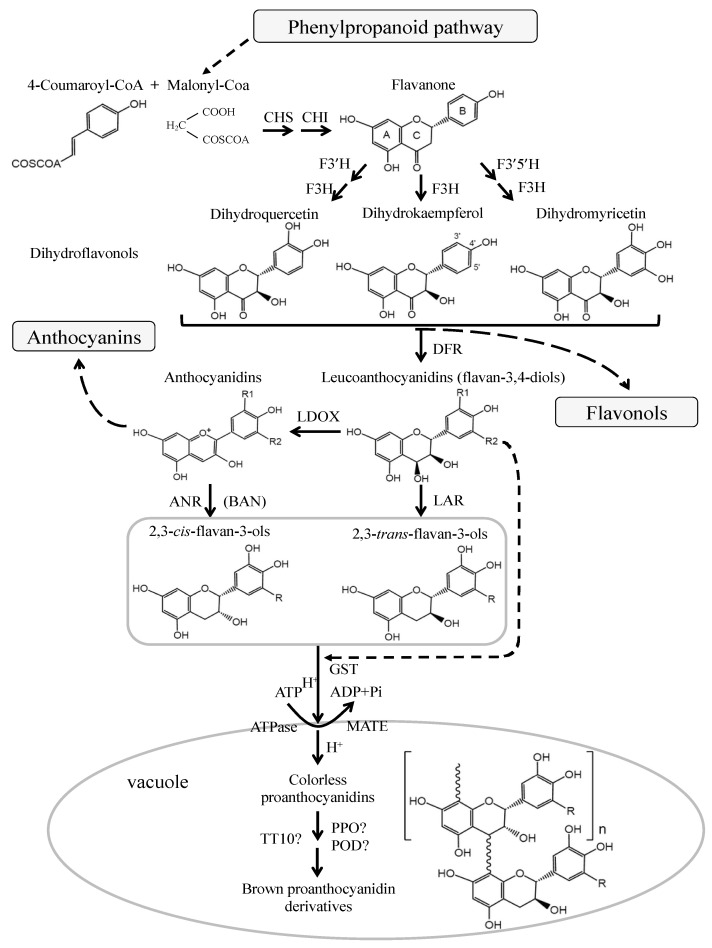
Biosynthetic pathway of flavonoid (adapt from [1,38]). The enzymes are: CHS, chalcone synthase; CHI, chalcone isomerase; F3H, flavanone 3-hydroxylase; F3′H, flavonoid 3′-hydroxylase; F3′5′H, flavonoid 3′,5′-hydroxylase; DFR, dihydroflavonol-4-reductase; LAR, leucoanthocyanidin reductase; ANS, anthocyanidin synthase; LDOX, leucoanthocyanidin dioxygenase; GST, glutathione S-transferase; MATE, multidrug and toxic compound extrusion; PPO, polyphenol oxidase; POD, peroxidase; TT10, transparent testa 10.

**Figure 3 foods-09-01774-f003:**
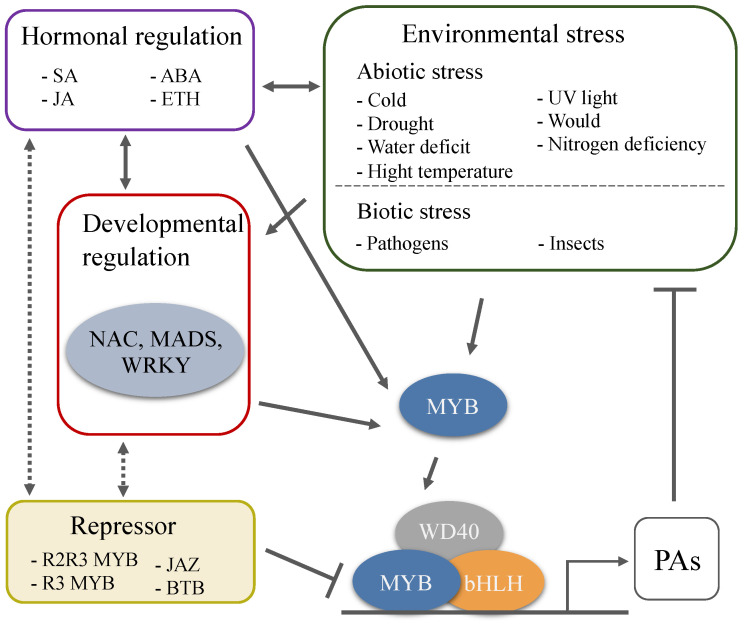
Environmental and developmental regulation of PAs biosynthesis through the MBW complexes. A large number of MYBs regulate the synthesis of PAs that are involved in plant response to stress and signals. Plant hormones are response to the environmental and developmental cues, and are the key factors involved in PAs biosynthesis. Environmental stresses affect the accumulation of PAs, and the high levels of PAs further enhance plant tolerance to extreme environmental challenges.

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
