# Peer review of "Advances in Biosynthesis and Biological Functions of Proanthocyanidins in Horticultural Plants"

_foods, 2020, doi:10.3390/foods9121774_

Round 1

Reviewer 1 Report

The authors reviewed the current status of knowledge on the regulation of key regulators on proanthocyanidins biosynthesis and involvement of proanthocyanidins in the resistance of plants against abiotic and biotic stress conditions. However, the authors should improve their review by performing a deep analysis of the available literature about plants that grow under stress and their chemical composition; the following papers are recommended:

Sobeh, M., Mahmoud, M. F., Abdelfattah, M. A., Cheng, H., El-Shazly, A. M., & Wink, M. (2018). A proanthocyanidin-rich extract from Cassia abbreviata exhibits antioxidant and hepatoprotective activities in vivo. Journal of ethnopharmacology213, 38-47.

Erasto, Paul, and R. T. Majinda. "Bioactive proanthocyanidins from the root bark of Cassia abbreviata." International Journal of Biological and Chemical Sciences 5.5 (2011): 2170-2179. Ramsay, A., & Mueller-Harvey, I. (2016). Senna alata leaves are a good source of propelargonidins. Natural Product Research30(13), 1548-1551.

Line 266, the authors claimed that planta with high levels of PAs may display an enhanced tolerance to extreme environmental challenges, please elaborate

Lines 269-277, the authors claimed that PAs are widely involved in plant resistance against the biotic stress, such as fungal pathogen infection and insect herbivores, please insert a reference

Reviewer 2 Report

Present ms adds to the numerous reviews published over the last years on the  importance,  genetics and environmental regulation of Proanthocyanidins (PAs) . PAs are plant  secondary metabolites also known as condensed tannins. Both the structure and organization of the ms sound well, but several sentences throughout the ms do not read well and need to be rephrased. In addition to this, at the light of the fact that present ms is submitted to “Foods” I would have expected to see a paragraph in which studies about the importance and effects of these metabolites for human/ and animal nutrition were reviewed. Conversely, these aspects are de facto overlooked here.   

Finally, there are several points that need to be clarified and better discussed and/or addressed. These points are listed below.  

Abstract

In face of what is stated “..and discussed the directions for future research on the metabolic  engineering of proanthocyanidins production to improve food and fruit crop quality” I found no any part in the ms in which these directions are throughfully discussed.

 Introduction

Line 48. WD40s  are regulatory proteins but their binding to promoters has not been proved yet, so they should not be regarded to as Transcription factors.

Line 68. “To date, LAR and ANR genes have been cloned from several plant species, and their functions  have been demonstrated both genetically and biochemically [30]”. More references about cloning and function of ANR and LAR genes should be added here. The activity of these enzymes from species spanning from Arabidopsis, sainfoin, grape to Lotus corniculatus have been shown earlier than [30], see for instance Xie et al. Science 2003; Tanner et al. J Biochem Chem 2003; Bogs et al. Plant Physiol 2005; Paolocci et al. Plant Physiol 2007.

Line 82. “VvMATE1 and VvMATE1 are two transporters ….” Please fix it

  1. Transcriptional activator of proanthocyanidins biosynthesis

The list of MYB activators in the species mentioned is incomplete, as an example in M. trunctula PAR is an additional activator of PAs. It should be also stated that the PA-related MYB activators are divided into three phylogenetic groups and the specificity of these members should be explained accordingly.

Lines 139-143. The ms would benefit from providing more details about the additional regulators cited in this paragraph. The readers will surely appreciate to know how, to which extent and under what  conditions MADS, bZIP, WRKY and NAC regulators interact with the MBW ternary complexes and/or directly with the promoters of structural genes of flavonoids

Line 148 “FaMYBF1” should be “FaMYB1”   

Line 153 “PhAN1” the family of this protein should be introduced

Lines 166-169. Please explain more in depth the passive and active repression function of MtMYB2 and PpMYB18.

Line 220-221. “Both regulated deficit irrigation and non-irrigated upregulated the 220 expression of PAs-related genes, but did not lead to changes in PAs concentration [102]”. Awkward sentence.

  1. Proanthocyanidins in response to biotic stresses

Here the authors should explain the role of PAs as scavengers of ROS better. Indeed PAs are accumulated in the vacuoles while ROS are produced in the plastids…

  1. 7. Hormones regulate proanthocyanidins synthesis

Line 360. “These results  suggest that SA is the primary hormone that control antifungal flavan-3-ols and PAs synthesis [139]”. This sentence is repeated twice see, please line 364.

Reviewer 3 Report

The work is well done. The only issues I see are that the linkages that are being made during what the authors call the condensation process are not well explained or the various complications that occur with those structures. Oddly there really aren't structures provided to explain what types of sub-units the polymers have. The authors also don't really distinguish the different types of sub-units carefully as they have neglected epigallocatechin and epicatechin gallate. The authors can clarify all of this with a figure containing the chemical structures. The authors do a good job of trying to explain a lot of complicated abbreviations throughout although after slogging through it I still found myself going back to remind myself about what some of the abbreviations meant. Since the paper is primarily about the regulatory mechanisms of PAs it might be good if they made a figure that summarizes their review. Below are a few select comments I have for the paper.

Abstract: Proanthocyanidins elicit astringency which is a tactile sensation not a flavor. This must be corrected. This is stated in line 29 and 30 in the manuscript so it unclear why this distinction isn’t obvious to the authors.

Line 34. What do you mean condensed? If you mean linked together through interflavan bonds you might as well go right ahead and say so. This is a review after all.

Line 38 drop the fruit ripening and just say fruit ripening. Actually many papers have evaluated the changes in grape skin proanthocyanidin development and there are multiple trends observed. The one you describe is certainly one example. In any case the authors should make distinctions about changes that occur in fruit that are as a result of biosynthesis or changes that occur due to changes in the size of fruit. This is a simple distinction. Regrettably your writing makes no clarification.

Line 86-94 is confusing. You don’t explain the relationship between why the seed coat is brown and proanthocyandins. It is unclear what you mean by rescue? Your subject verb usage is also troubling here which possibly explains the lack of clarity. Proanthocyanidins turn brown when oxidized but so do most phenolics. Proanthocyanidins comparatively are weak substrates for oxidative enzymes as compared to lower molecular weight phenolics though the brown polymers they make are densely colored. 

Round 2

Reviewer 1 Report

The authors answered the raised comments 

Author Response

Response: We truly thank you for your comments, which have greatly improved the integrity of our work.

Reviewer 2 Report

The authors have addressed most of the concerns raised in my previous revision.

In particular, I appreciate the addition of new comments, references, and figures. This review recapitulates the most interesting findings concerning the genetic and environmental regulation of PA biosynthesis and accumulation and the role played by these metabolites in plant stress response. Unfortunately, however, what has not been done during the revision it is the careful check of its style and language. As a matter of fact, it remains plenty of sentences that do not read well. The use of the verbs is also not always consistent throughout the ms.

In addition to the points above, I guess that part (4) of fig 1 should be Procyanidin B1 and no A1, and I suggest adding “secondary metabolites beneficial to human health” at the end of the first sentence of the Introduction section. Finally, the name of the genes should always be given in italics.

Sentences to be rephrased

Lines 141-145: to be rephrased

Line 162: “Complex” instead of “complexes”

Line 193: “MBW is not the only TFs….” MBW is indeed a complex and not a TF

Lines 202-203: to be rephrased

Line 217: As above. Here also delete “respectively” and divide the sentence into two sentences

Line 227-229: to be rephrased

Line 247-248: as above

Line 296-297: as above

Line 312-315: as above

Line 335-337: as above

Line 343-345 as above

Lie 356: delete “that”

Line 375: to be rephrased

Line 384: the verb is not correct

Line 401: replace “have“ with “are“

Line 402: delete “that”

Lines 410- 413: to be rephrased

Lines 429-430 and 434-435: duplicate sentence

Lines 462 -463: “leads” and “brings”

Line 466-467: to be rephrased

Author Response

This manuscript is a resubmission of an earlier submission. The following is a list of the peer review reports and author responses from that submission.